# Empagliflozin reduces podocyte lipotoxicity in experimental Alport syndrome

Mengyuan Ge[1,2], Judith Molina[1,2], Jin-Ju Kim[1,2], Shamroop K Mallela[1,2], Anis Ahmad[3], Javier Varona Santos[1,2], Hassan Al-Ali[1,2], Alla Mitrofanova[1,2], Kumar Sharma[4], Flavia Fontanesi[5], Sandra Merscher[1,2], Alessia Fornoni[1,2]*

[1]Katz Family Division of Nephrology and Hypertension, Department of Medicine, University of Miami Miller School of Medicine, Miami, United States; [2]Peggy and Harold Katz Family Drug Discovery Center, University of Miami Miller School of Medicine, Miami, United States; [3]Department of Radiation Oncology, University of Miami Miller School of Medicine, Miami, United States; [4]Center for Precision Medicine, School of Medicine, University of Texas Health San Antonio, San Antonio, United States; [5]Department of Biochemistry and Molecular Biology, University of Miami, Miami, United States

**Abstract** Sodium-glucose cotransporter-2 inhibitors (SGLT2i) are anti-hyperglycemic agents that prevent glucose reabsorption in proximal tubular cells. SGLT2i improves renal outcomes in both diabetic and non-diabetic patients, indicating it may have beneficial effects beyond glycemic control. Here, we demonstrate that SGLT2i affects energy metabolism and podocyte lipotoxicity in experimental Alport syndrome (AS). In vitro, we found that the SGLT2 protein was expressed in human and mouse podocytes to a similar extent in tubular cells. Newly established immortalized podocytes from Col4a3 knockout mice (AS podocytes) accumulate lipid droplets along with increased apoptosis when compared to wild-type podocytes. Treatment with SGLT2i empagliflozin reduces lipid droplet accumulation and apoptosis in AS podocytes. Empagliflozin inhibits the utilization of glucose/pyruvate as a metabolic substrate in AS podocytes but not in AS tubular cells. In vivo, we demonstrate that empagliflozin reduces albuminuria and prolongs the survival of AS mice. Empagliflozin-treated AS mice show decreased serum blood urea nitrogen and creatinine levels in association with reduced triglyceride and cholesterol ester content in kidney cortices when compared to AS mice. Lipid accumulation in kidney cortices correlates with a decline in renal function. In summary, empagliflozin reduces podocyte lipotoxicity and improves kidney function in experimental AS in association with the energy substrates switch from glucose to fatty acids in podocytes.

## Editor's evaluation

The article has important scientific merit in the field of glomerular diseases. The authors propose a link between inhibition of SGLT2 and lipotoxicity-mediated renal injury in experimental Alport syndrome (AS) by modulation pathways linked to CKD progression, possibly through metabolic adaption in podocytes.

## Introduction

AS is a hereditary disease of glomerular basement membranes caused by mutations in collagen type IV genes A3, A4, and A5 (*Barker et al., 1990*; *Gross et al., 2016*; *Longo et al., 2002*). AS is characterized by renal fibrosis with progression to end-stage renal disease in young adult life (*Barker et al.,*

*For correspondence: AFornoni@med.miami.edu

*1990*; *Grünfeld, 2000*; *Williamson, 1961*). Though early treatment with angiotensin-converting enzyme inhibitors (ACEi) was shown to reduce proteinuria and delay disease progression in both retrospective (*Gross et al., 2012*) and prospective (*Boeckhaus et al., 2022*) studies, there is no specific treatment to prevent renal failure in patients with AS.

SGLT2i, initially developed for the treatment of patients with type 2 diabetes (T2D), were recently found to protect from kidney and cardiovascular outcomes in both diabetic and non-diabetic patients with chronic kidney disease (CKD) (*Heerspink et al., 2020*). SGLT2 is most abundantly expressed in the apical brush border membrane of the proximal tubule, where it plays a key role in renal glucose reabsorption (*Vallon et al., 2011*). SGLT2i selectively blocks SGLT2, thereby enhancing urinary glucose excretion and reducing glycemia (*DeFronzo et al., 2017*; *Novikov and Vallon, 2016*; *Vallon and Thomson, 2017*). While the major mechanism for renoprotection is thought to involve the tubuloglomerular feedback and glomerular hemodynamics (*Cherney et al., 2014*), SGLT2i may also modulate key metabolic pathways linked to CKD progression. In response to increased glycosuria, the body engenders a metabolic adaption to enhance the usage of fat for energy production (*Ferrannini et al., 2016*). Additional studies have also shown that SGLT2i enhances β-oxidation in the liver (*Wallenius et al., 2022*) and improves liver fat deposition in patients with T2D and fatty liver disease (*Kuchay et al., 2018*; *Shibuya et al., 2018*). Similarly, SGLT2i lowers the cardiac content of cardiotoxic lipids in obese diabetic rats (*Aragón-Herrera et al., 2019*). These observations suggest a possible link between SGLT2i and lipid metabolism. We and others have demonstrated that the accumulation of both cholesterol esters and fatty acids in podocytes and tubular cells contributes to the pathogenesis of AS (*Ding et al., 2018*; *Kim et al., 2021*; *Mitrofanova et al., 2018*; *Wright et al., 2021*), indicating that reducing the lipid content in the kidney may potentially reduce lipotoxicity-mediated renal injury in AS.

Although all cells in the kidney are high energy-demanding, the metabolic substrates for ATP production are cell type-dependent (*Console et al., 2020*). Renal proximal tubular cells in particular use free fatty acids as the preferred fuel, whereas inhibition of fatty acid oxidation (FAO) renders tubular cells susceptible to cell death and lipid accumulation (*Kang et al., 2015*). Podocytes usually rely on glucose for energy production, while fatty acids are used as an alternative substrate (*Abe et al., 2010*; *Brinkkoetter et al., 2019*). Interestingly, a recent study demonstrated SGLT2 expression in podocytes and its expression was modulated by exposure to albumin, although the functional relevance of SGLT2 expression in podocytes is unknown (*Cassis et al., 2018*). With this study, we aimed at investigating if SGLT2i affects energy metabolism in both podocytes and tubular cells in experimental AS.

## Results

### SGLT2 is expressed in human podocytes and immortalized podocytes established from wild-type (WT) and AS mice

Immunohistochemistry in normal human kidney sections demonstrated both glomerular and proximal tubule expression of SGLT2 (*Figure 1A*). Using western blot analysis, we demonstrate similar levels of SGLT2 protein expression levels in cultured human podocytes when compared to HK2 tubular cells. Mouse liver lysate, HepG2 liver cancer cells, and kidney lysate from *Sglt2*[-/-] mouse were used as the negative controls (*Figure 1B*). To study the effect of SGLT2i in an experimental model of non-diabetic kidney disease, we developed immortalized podocytes and tubular cell lines established from SV40[+]; *Col4a3*[+/+] (immorto-WT) and SV40[+]; *Col4a3*[-/-] (immorto-AS) mice. The expression of the podocyte-specific marker Synaptopodin (SYNPO) and of the tubule-specific marker Aquaporin 1 (AQP1) was confirmed in podocyte and tubular cell lines, respectively (*Figure 1—figure supplement 1*). We found similar SGLT2 protein expression levels in both tubular cells and podocytes (*Figure 1C and D*), while *Sglt2* mRNA expression levels were significantly higher in AS tubular cells and podocytes than in WT controls (*Figure 1E*). To confirm glomerular expression of SGLT2 in vivo, kidney cortices of Alport mice were co-stained with SYNPO and SGLT2 antibodies. As expected, the colocalization of SYNPO with SGLT2 staining confirmed that SGLT2 is expressed in podocytes (*Figure 1F*). Additionally, we examined the Kidney Interactive Transcriptomics (https://humphreyslab.com/SingleCell/) which is an online analysis tool for kidney single-cell datasets (*Wu et al., 2018*). In the Healthy Mouse Dataset

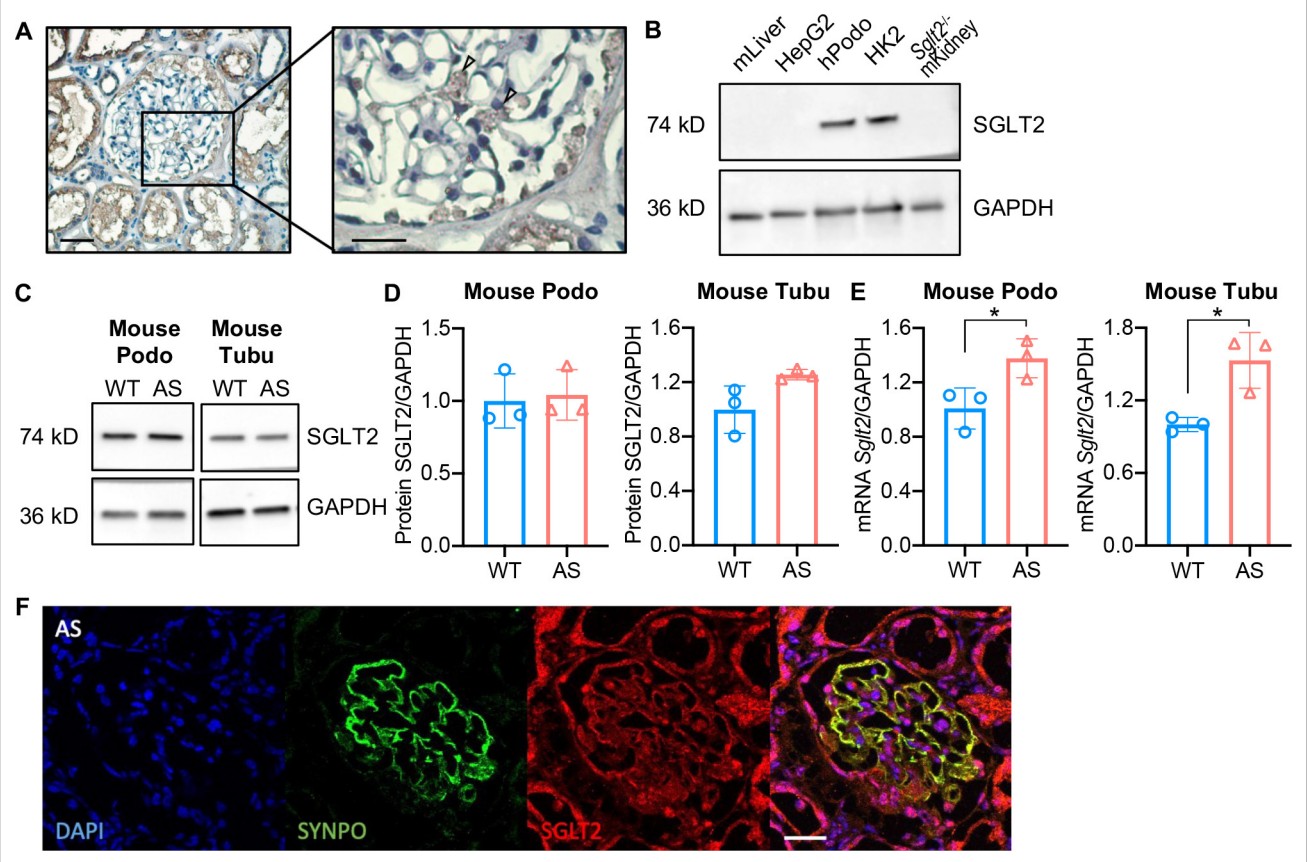

**Figure 1.** Sodium-glucose cotransporter-2 (SGLT2) protein is expressed in the human kidney cortex and in cultured human and mouse podocytes. (**A**) Immunohistochemistry staining of human kidney cortex for SGLT2 (left panel, scale bar: 50 µm; right panel, scale bar: 25 µm). (**B**) Western blot images demonstrating SGLT2 expression in cultured human podocytes (hPodo). Mouse liver lysate (mLiver), HepG2 liver cancer cells, and kidney lysate from *Sglt2*[-/-] mouse (*Sglt2*[-/-] mKidney) were used as the negative controls. HK2 proximal tubular cells were used as the positive control. (**C,D**) Western blot images (**C**) and quantification (**D**) demonstrating SGLT2 expression in mouse proximal tubular cells (Tubu) and podocytes (Podo) established from wild-type (WT) and Alport (AS) mice (n=3). (**E**) *Sglt2* mRNA expression in WT and AS podocytes and tubular cells (n=3). (**F**) Representative confocal images of kidney cortices of AS mice (scale bars: 25 µm) stained with DAPI (blue), Synaptopodin (SYNPO, green) and SGLT2 (red). Yellow represents the co-localization of SYNPO and SGLT2. (**D**), (**E**), Two-tailed Student's t-test. *p<0.5.

The online version of this article includes the following source data and figure supplement(s) for figure 1:

**Source data 1.** Uncropped western blot images of *Figure 1*.

**Figure supplement 1.** Podocyte-specific marker Synaptopodin (SYNPO) and tubule-specific marker Aquaporin 1 (AQP1) were confirmed in podocyte and tubular cell lines, respectively.

**Figure supplement 1—source data 1.** Uncropped western blot images of *Figure 1—figure supplement 1*.

**Figure supplement 2.** Single-cell transcriptomics indicates the expression of *Sglt2* in podocyte.

---

(***Wu et al., 2019***), *Sglt2* is expressed in podocytes, though its expression level in every cell type is low (***Figure 1—figure supplement 2***).

## Treatment of AS podocytes with empagliflozin reduces lipid droplet accumulation and apoptosis

We previously described that AS podocytes are characterized by increased apoptosis and lipid droplet (LD) accumulation when compared to WT podocytes (***Kim et al., 2021***; ***Liu et al., 2020***). To further evaluate whether SGLT2i can reduce lipotoxicity in podocytes as well as in tubular cells isolated from AS mice, immortalized WT and AS podocytes and tubular cells were treated with empagliflozin or vehicle. SGLT2i significantly decreases cytotoxicity in empagliflozin-treated compared with vehicle-treated AS tubular cells (***Figure 2A***). No differences in apoptosis and lipid droplet accumulation were observed in any of the groups (***Figure 2B and E***). As expected, AS podocytes showed increased

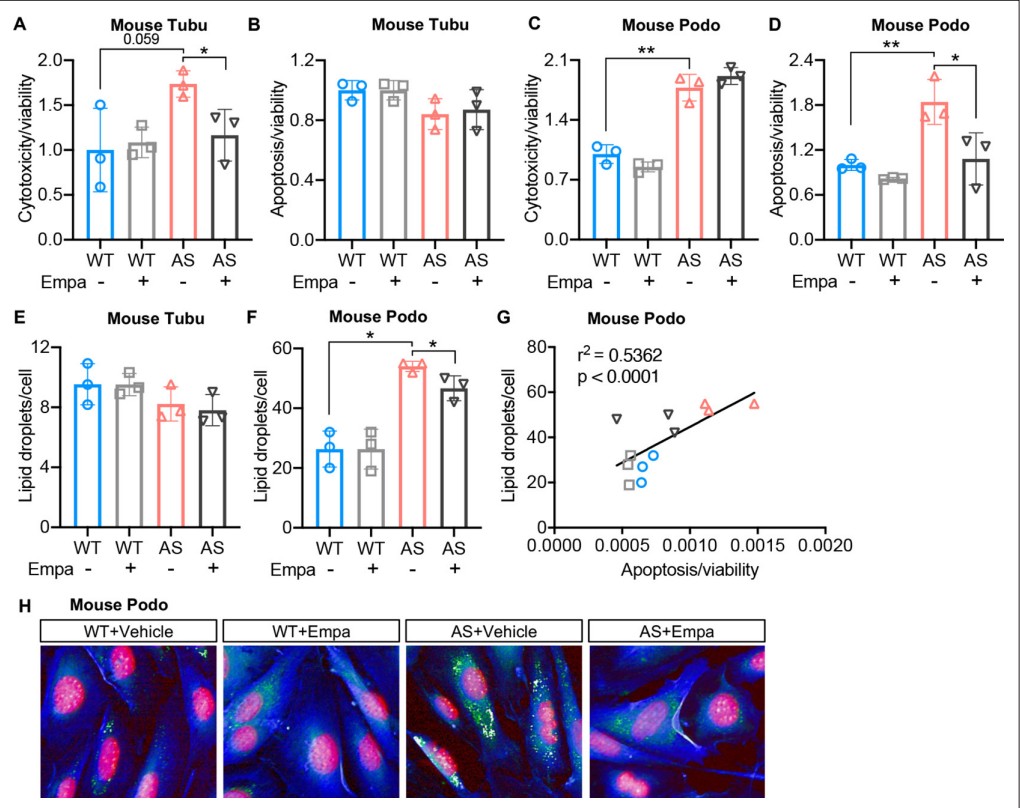

**Figure 2.** Treatment of Alport syndrome (AS) podocytes with empagliflozin reduces lipid droplet (LD) accumulation and apoptosis. (**A–D**) Immortalized podocytes and tubular cells of wild-type (WT) and AS mice treated with empagliflozin (Empa) or vehicle for 48 hr. (**A,C**) Bar graph analysis showing cytotoxicity normalized to viability, then compared to WT (n=3). (**B,D**) Bar graph analysis showing apoptosis normalized to viability, then compared to WT (n=3). (**E,F**) LD accumulation in tubular cells (**E**) and podocytes (**F**) was measured by Nile red staining. Bar graph analysis showing the quantification of the number of LDs per cell (n=3). (**G**) Correlation analyses between the LD accumulation and apoptosis in podocytes (n=12). (**H**) Representative images of Nile red staining demonstrate increased LD numbers (Nile red: green) in AS podocytes (Cell mask blue: blue; DAPI: red) compared to WT podocytes, which is reduced by Empa treatment. (**A–F**), Two-tailed Student's t-test. (**G**), Pearson's correlation coefficient. *p<0.5, **p<0.01.

The online version of this article includes the following figure supplement(s) for figure 2:

**Figure supplement 1.** Bar graph analysis of glucose content in Alport syndrome (AS) podocyte with or without empagliflozin (empa) treatment (n=3).

cytotoxicity, apoptosis, and intracellular LD when compared to WT podocytes (*Figure 2C, D and F*). Empagliflozin treatment significantly reduced apoptosis and intracellular LD, but not cytotoxicity in AS podocytes (*Figure 2C, D and F*). Representative picture of Nile red staining revealed fewer LD per cell in empagliflozin-treated compared with vehicle-treated AS podocytes (*Figure 2H*), suggesting empagliflozin ameliorates lipotoxicity in AS podocytes. Interestingly, we observed a positive correlation between LD accumulation and apoptosis (*Figure 2G*). Treatment with empagliflozin significantly reduced the glucose content in both podocytes and tubular cells established from AS mice, which may result in glucose deprivation and a shift in energy fuel (*Figure 2—figure supplement 1A and B*).

## Empagliflozin inhibits the utilization of pyruvate as a metabolic substrate in AS podocytes

To investigate if empagliflozin affects metabolic substrate preferences, endogenous cellular and coupled substrate-driven respiration were measured by high-resolution respirometry. Endogenous respiration measured in intact cells was not altered in either AS tubular cells or podocytes compared to WT (*Figure 3A and B*). Cells were then permeabilized with digitonin and substrates for fatty

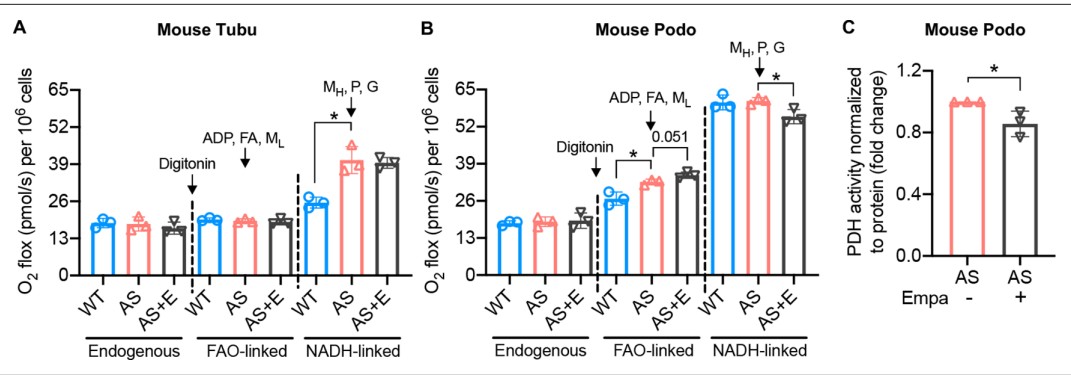

**Figure 3.** Empagliflozin inhibits the utilization of pyruvate as a metabolic substrate in Alport syndrome (AS) podocytes. (**A, B**) Bar graph analysis of endogenous and substrate-driven oxygen consumption rates in wild-type (WT) and Alport (AS) tubular cells (**A**) and podocytes (**B**) treated with or without empagliflozin (**E**) (n=3). The sequential addition of permeabilizing agent and substrates was labeled in the figure. (**C**) Pyruvate dehydrogenase (PDH) activity was measured by a colorimetric assay in protein extracts from AS podocytes, normalized to protein concentration (n=3). Two-tailed Student's t-test, *p<0.5. FA: octanoylcarnitine; $M_L$: malate-low concentration; $M_H$: malate-high concentration; P: pyruvate; G: glutamate.

The online version of this article includes the following figure supplement(s) for figure 3:

**Figure supplement 1.** Empagliflozin inhibits NADH-linked oxygen consumption rate in Alport syndrome (AS) podocytes.

acids-driven and nicotinamide adenine dinucleotide (NAD)-driven respiration were provided sequentially. No difference in oxygen consumption rate (OCR) was detected between WT and AS tubular cells in response to fatty acids. However, AS tubular cells show elevated respiration after the addition of NADH-linked substrates. Treatment of empagliflozin did not affect the respiration of AS tubular cells independently of the substrate (**Figure 3A**). In contrast to tubular cells, AS podocytes showed a slightly but significant increase in FAO-linked OCR compared to WT podocytes, which could be due to the increase in intracellular lipid accumulation. This increase was maintained upon empagliflozin treatment and showed a tendency to increase, though not significant (**Figure 3B**). Moreover, the addition of NADH-linked substrates to WT and AS podocytes increased OCR to approximately the double of the value recorded in the presence of FAO-linked substrates, in agreement with podocytes preferential use of glucose oxidation for ATP production. Interestingly, NADH-linked respiration in AS podocytes was inhibited by treatment with empagliflozin (**Figure 3B**). To confirm the inhibitory effect of empagliflozin on NADH-driven respiration, we repeated the assay by measuring directly NADH-driven respiration without the addition of fatty acids. A similar change was observed (**Figure 3—figure supplement 1A and B**). Taken together, these data suggest that in podocytes established from AS mice, empagliflozin may induce a metabolic remodeling characterized by a reduction in glucose oxidation and a switch toward the use of alternative substrates for ATP production. To further characterize the adaptation to energy sources in AS podocytes, pyruvate dehydrogenase (PDH) activity was measured. PDH is an enzyme that converts glycolysis-derived pyruvate to acetyl-CoA and increases its influx into the tricarboxylic acid (TCA) cycle (**Zhang et al., 2014**). PDH plays a central role in the reciprocal regulation of glucose and lipid oxidation (**Zhang et al., 2014**). We found that PDH activity was reduced in AS podocytes by empagliflozin treatment (**Figure 3C**), suggesting a switch to the consumption of fatty acids as energy fuel. Additionally, we found that empagliflozin treatment reduced glycolysis in AS podocytes (**Figure 3—figure supplement 1C**). This finding is similar to what was observed in diabetic kidneys (**Li et al., 2020**) where aberrant glycolysis was inhibited by empagliflozin.

## *Sglt2* knockdown reduces lipotoxicity in AS podocytes

To confirm the anti-lipotoxic effects of SGLT2i in podocytes, we used gene silencing of SGLT2 by siRNA transfection in AS podocytes. We show that *Sglt2* siRNA transfection downregulates the expression of *Sglt2* in AS podocytes (**Figure 4A and B**). AS podocytes transfected with a nontargeting siRNA control (siCtrl) and *Sglt2* siRNA (siSglt2) were treated with empagliflozin or vehicle. Cytotoxicity and apoptosis of AS podocytes were analyzed in siCtrl, siCtrl + empagliflozin, siSglt2, and siSglt2 +

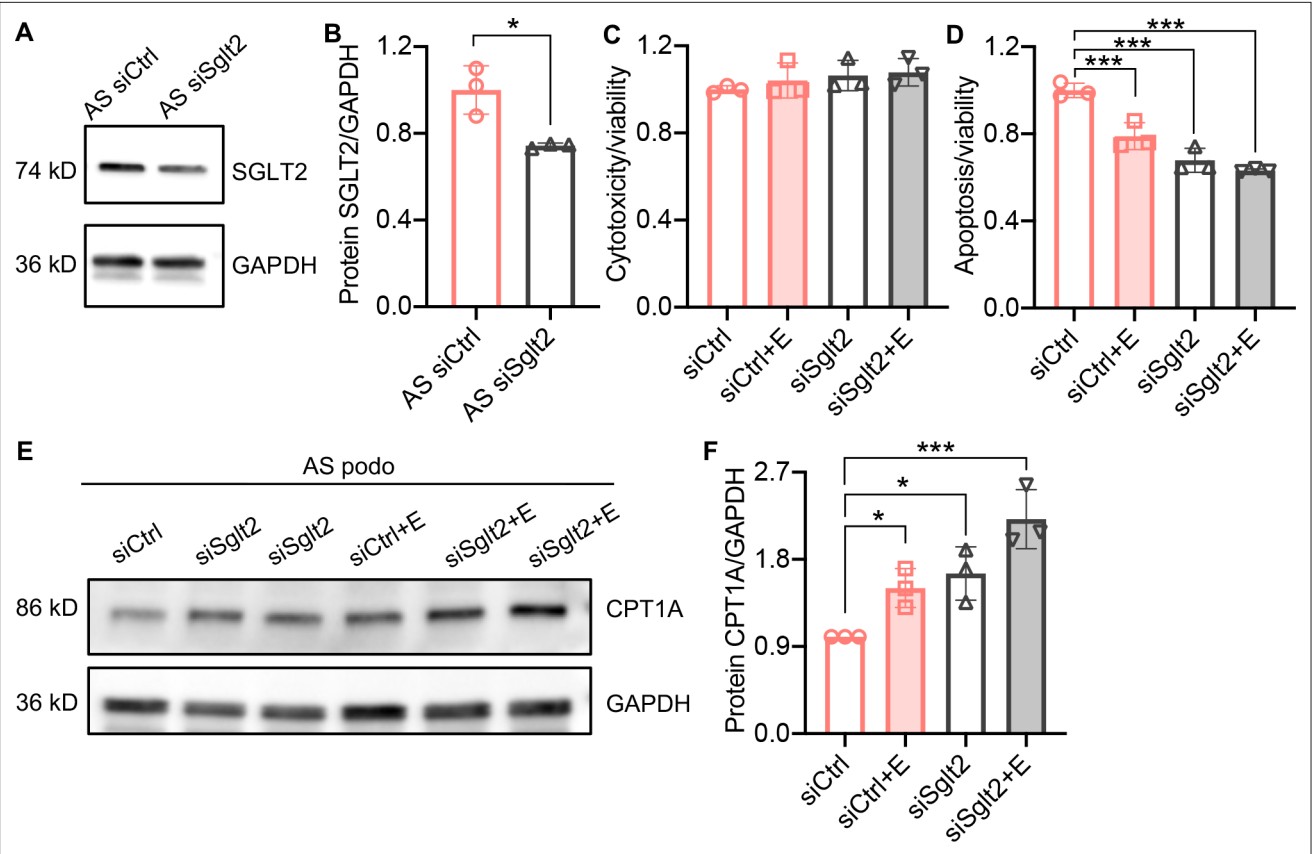

**Figure 4.** *Sglt2* knockdown reduces lipotoxicity in Alport syndrome (AS) podocytes. (**A, B**) Western blot images (**A**) and quantification (**B**) of Sodium-glucose cotransporter-2 (SGLT2) protein in AS podocytes transfected with *Sglt2* siRNA (siSglt2) or nontargeting siRNA (siCtrl) for 72 hr. GAPDH was used as a sample loading control (n=3). (**C,D**) Bar graph analysis showing cytotoxicity (**C**) and apoptosis (**D**) normalized to viability (n=3) in siCtrl and siSGLT2 AS podocytes, with or without the treatment of empagliflozin (**E**), then compared with siCtrl. (**E,F**) Western blot images (**E**) and quantification (**F**) of CPT1A protein in siCtrl and siSglt2 AS podocytes, with or without the treatment of empagliflozin (n=3). (**B**), Two-tailed Student's t-test, (**C**), (**D**), (**F**), One-Way ANOVA followed by Holm-Sidak's multiple comparisons. *p < 0.05, ***p < 0.001.

The online version of this article includes the following source data for figure 4:

**Source data 1.** Uncropped western blot images of *Figure 4*.

empagliflozin groups. Similar to siCtrl + empagliflozin AS podocytes, siSglt2, and siSglt2 + empagliflozin AS podocytes showed reduced apoptosis compared with vehicle-treated siCtrl AS podocytes in the absence of changes in cytotoxicity (*Figure 4C and D*). To study FAO in siSglt2 AS podocytes, carnitine palmitoyltransferase 1 A (CPT1A) was determined by western blot analysis. CPT1A is the rate-limiting enzyme of FAO (*Schlaepfer and Joshi, 2020*), which was found upregulated by empagliflozin treatment and knockdown of *Sglt2*. This observation is consistent with an interventional clinical trial in which dapagliflozin treatment led to an adaptive preference of skeletal muscle metabolism for fatty acids, as evidenced by increased expression of CPT1A (*op den Kamp et al., 2022*).

## Empagliflozin prolongs the survival of AS mice

To investigate if empagliflozin can improve survival in mice with non-diabetic renal disease which typically die from renal failure, AS mice were fed an empagliflozin-supplemented chow (70 mg/kg) or a regular diet starting at 4 weeks of age for 6 weeks. Mice with experimental AS start developing proteinuria at 4 weeks of age, followed by death at 8–9 weeks of age. We found that empagliflozin extended the lifespan of AS mice by about 22% compared to untreated AS mice (*Figure 5A*). Blood glucose was measured at 8 weeks of age and no difference was observed in empagliflozin-treated compared to untreated AS mice (*Figure 5B*). These data suggest that the ability of empagliflozin to prolong the survival of AS mice is independent of its anti-hyperglycemic effects.

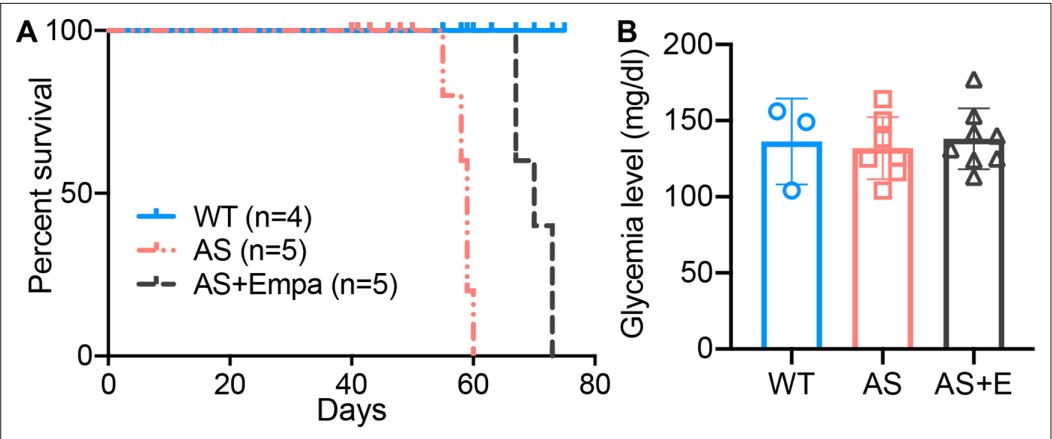

**Figure 5.** Empa improves the survival of Alport syndrome (AS) mice. (**A**) Survival curve (n=4–5) of AS mice fed empagliflozin-supplemented (**E**) chow versus placebo diet starting at 4 weeks of age, compared to age-matched wild-type (WT) control mice. (**B**) Glycemia levels of WT and AS mice fed placebo diet and AS mice fed empagliflozin chow (n=3–8). (**B**), AS vs AS +E: Two-tailed Student's t-test.

## Empagliflozin improves renal function in a mouse model of Alport syndrome

To study the effects of SGLT2 inhibitors on the renal outcome, AS mice were fed empagliflozin-supplemented chow starting at 4 weeks of age for 4 weeks, and the renal phenotype was compared to AS mice fed a regular diet. Ramipril, an ACEi used as a standard of care for patients with AS, was also used alone or in combination with empagliflozin. At 4 weeks of age, ramipril was added to the drinking water and/or mice were fed with an empagliflozin-supplemented chow as indicated. Mice on the different regimens were compared with AS mice fed a regular diet. All mice were sacrificed at 8 week of age. Empagliflozin, ramipril, and the empagliflozin + ramipril (E+R) combination significantly reduced the albumin-to-creatinine ratio (ACR) and prevented body weight loss in AS mice (**Figure 6A and B**). Empagliflozin, ramipril, and E+R significantly reduced blood urea nitrogen (BUN) and creatinine levels in AS mice (**Figure 6C and D**). Unlike what has been observed in patients enrolled in DAPA-CKD, the addition of empagliflozin to the standard of care (SOC) ramipril did not confer additional renoprotection, and overall, no difference across treatment groups was observed. Glomeruli of AS mice exhibited significant mesangial matrix expansion (**Figure 6E**) as determined by Periodic acid-Schiff (PAS) staining and significantly increased fibrosis as determined by Picrosirius red staining (**Figure 6F**), which were reduced by the treatment of empagliflozin, ramipril, and E+R. Empagliflozin, ramipril, and E+R treatment of AS mice also prevented podocyte loss as suggested by similar podocyte numbers, as indicated by increased Wilms tumor 1 (WT1)-positive cells per glomerulus, in treated AS compared to WT mice (**Figure 6G**).

## Empagliflozin prevents renal lipid accumulation in experimental Alport syndrome

To investigate whether empagliflozin prevents lipid accumulation in kidney cortices of AS mice, Oil Red O (ORO) staining was performed. We found an increased number of LD-positive glomeruli in AS mice, while the number of LD-positive glomeruli in all treatment groups was similar to WT mice (**Figure 7A**). We then extracted lipids from kidney cortices to investigate the composition of specific lipids and found increased cholesterol ester (CE) and triglyceride contents in AS compared to WT mice (**Figure 7B and D**), similar to what we previously reported (**Kim et al., 2021**). Interestingly, though all treatment groups showed a decreased CE content in kidney cortices, only empagliflozin and E+R reduced triglyceride levels. The total cholesterol content was similar in all five groups (**Figure 7C**). We previously demonstrated a correlation between lipid accumulation and renal function decline in experimental models of metabolic and non-metabolic kidney disease (**Ducasa et al., 2019**; **Ge et al., 2021**; **Wright et al., 2021**). Similarly, we found a positive correlation between the CE, and triglyceride content in kidney cortices with ACR, serum BUN, and creatinine levels (**Figure 7E–J**).

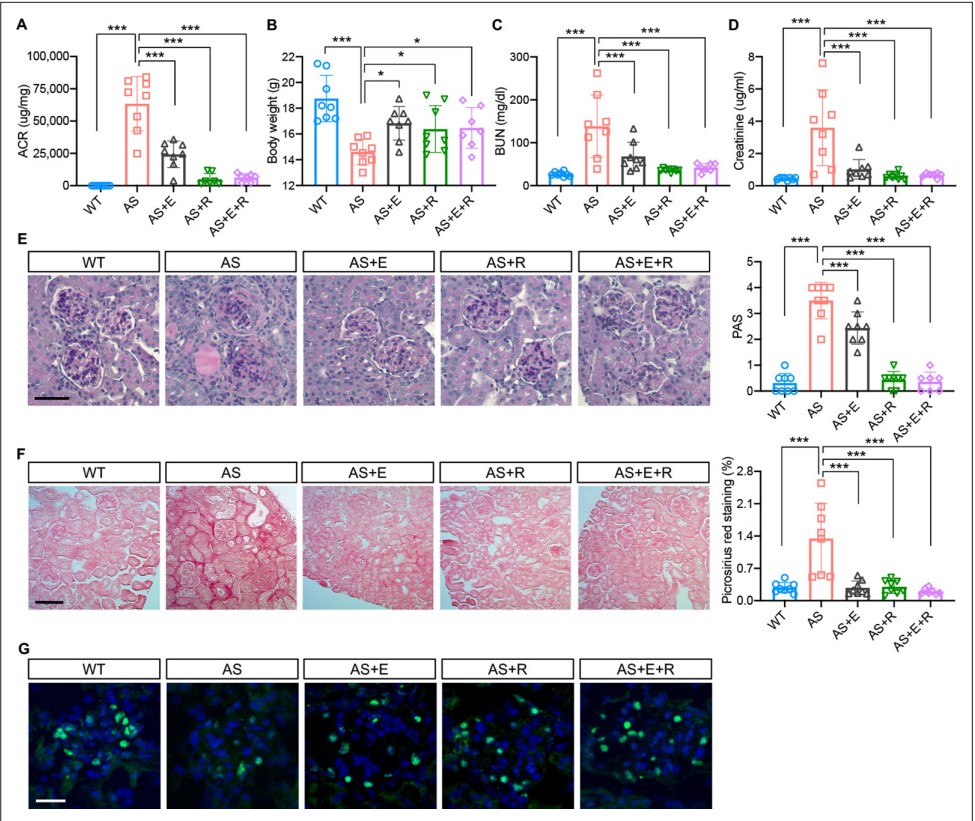

**Figure 6.** Empagliflozin improves renal function in a mouse model of Alport syndrome. (**A**) Urinary albumin-to-creatinine ratio (ACR) in WT and Alport syndrome (AS) mice fed with placebo, empagliflozin (**E**), ramipril (**R**), or the combination of empagliflozin and ramipril (E+R). Urines were collected at the time of sacrifice (n = 7–8). (**B**) Bar graph analysis of body weights of mice from all experimental groups. (**C,D**) Bar graph analysis of blood urea nitrogen (BUN) (**C**) and creatinine (**D**) levels of mice from all experimental groups (n = 7–8). (**E**) Representative images of Periodic acid-Schiff (PAS) staining and bar graph analysis showing the mesangial expansion score of kidney cortices sections (scale bar: 50 μm; n = 7–8). (**F**) Representative Picrosirius red staining and bar graph analysis showing the quantification of fibrosis in kidney cortices sections (scale bar: 100 μm; n=7–8). (**G**) Representative images of kidney cortices stained with WT1 (green) to detect podocytes and DAPI (blue) to reveal nuclei and bar graph quantification of the average number of WT1-positive podocytes per glomerulus (scale bar: 25 μm, n = 7–8). One-Way ANOVA followed by Holm-Sidak's multiple comparisons. *p < 0.05, **$P$p < 0.01, ***p < 0.001.

## Discussion

In the present study, we investigate the mechanisms by which empagliflozin, an SGLT2i, affects the usage of glucose and fatty acids as energy substrates in podocytes and tubular cells as well as the effects of empagliflozin on lipotoxicity-induced cell injury and renal function decline. Though SGLT2 is typically expressed in proximal tubules, its expression in glomerular mesangial cells (*Maki et al., 2019*; *Wakisaka et al., 2016*) and podocytes (*Cassis et al., 2018*) has been previously reported, suggesting that these cells can be potential targets for SGLT2i. Despite the absence of SGLT2 expression in glomerular endothelial cells, empagliflozin shows a protective effect on the glomerular endothelium via podocyte-endothelial cell crosstalk (*Locatelli et al., 2022*). SGLT2 expression in mesangial cells and podocytes is increased under high-glucose (*Wakisaka et al., 2016*) or protein-overload conditions (*Cassis et al., 2018*), respectively. In this study, we focused on investigating and comparing the lipid-modifying effects of empagliflozin in tubular cells and podocytes in experimental AS. Similarly, we demonstrate that the SGLT2 protein is expressed in the human kidney cortex, cultured human podocytes, as well as healthy and diseased mouse podocytes (*Figure 1*). *Sglt2* mRNA is increased in immortalized podocytes established from AS mice, but not at protein levels. We also show that SGLT2i reduces LD accumulation (*Figure 2*) and glucose/pyruvate-driven respiration (*Figure 3*) in AS podocytes. Similarly, the knockdown of *Sglt2* reduces lipotoxicity-mediated injury of AS podocytes

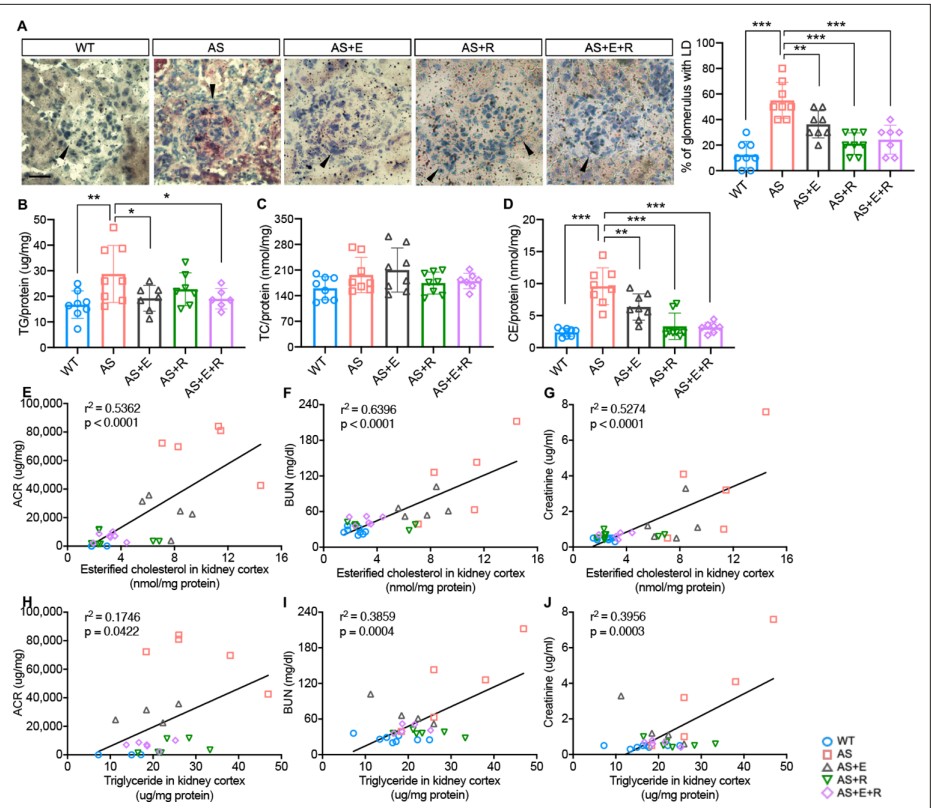

**Figure 7.** Empagliflozin prevents renal lipid accumulation in experimental Alport syndrome. (**A**) Representative Oil Red-O (ORO) images of stained kidney cortices sections (scale bar: 20 μm) and bar graph quantification of the number of glomeruli with lipid droplets (LD) in ORO-stained slides (n = 7–8). (**B–D**) Bar graph analysis of triglyceride TG, (**B**), total cholesterol TC, (**C**), and cholesterol ester CE, (**D**) contents in kidney cortices. Values are normalized to protein concentrations (n = 6–8). (**E–G**) Correlation analyses between the CE content of kidney cortices and albumin-to-creatinine ratio (ACR), blood urea nitrogen (BUN), or creatinine (n = 27, 31, 31). (**H–J**) Correlation analyses between the TG content of kidney cortices and ACR, BUN, or creatinine (n = 29, 29, 29). (**A–D**), One-Way ANOVA followed by Holm-Sidak's multiple comparisons. (**E–J**), Pearson's correlation coefficient. *p < 0.05, **p < 0.01, ***p < 0.001.

(*Figure 4*). In vivo, we demonstrate for the first time that empagliflozin prolongs the survival (*Figure 5*), reduces renal lipotoxicity, and prevents kidney disease progression (*Figures 6 and 7*) in an experimental model of AS.

Renal lipotoxicity contributes to the pathogenesis of several forms of kidney disease (*Ducasa et al., 2019*; *Pedigo et al., 2016*; *Yoo et al., 2015*). We previously demonstrated that impaired cholesterol efflux in podocytes plays a critical pathogenic role in diabetic kidney disease (DKD) (*Ducasa et al., 2019*; *Merscher-Gomez et al., 2013*) as well as in diseases of non-metabolic origin, including AS (*Kim et al., 2021*; *Mitrofanova et al., 2018*), where we have also observed altered free fatty acids metabolism. Others have reported that defective FAO is associated with lipid deposition and fibrosis in kidney tubules (*Kang et al., 2015*), and contributes to disease progression in AS (*Ding et al., 2018*). In this study, we utilized immortalized podocytes and tubular cells newly established from AS and WT mice. Proximal tubular cells are a known target of SGLT2i. Here, we aimed at investigating if podocytes and tubular cells change their preferences with regard to their metabolic fuel in response to SGLT2i. In the kidney, podocytes highly rely on glucose as the substrate for ATP production (*Abe et al., 2010*), while tubular cells use free fatty acid as the preferred energy source (*Kang et al., 2015*). Therefore, a reduction of glucose availability by SGLT2i may trigger the utilization of alternative energy substrates, such as lipids (*Osataphan et al., 2019*). To study the effect of SGLT2i on energy substrate switch, we first investigated whether SGLT2i exercises a protective effect on immortalized podocytes and tubular cells derived from AS mice. We demonstrate that AS podocytes have increased cytotoxicity and apoptosis when compared to WT podocytes (*Figure 2*). Empagliflozin treatment protected AS

podocytes from apoptosis but not cytotoxicity. On the other hand, AS tubular cells did not show increased apoptosis but were found to exhibit a tendency to increase cytotoxicity when compared to WT tubular cells. Tubular cytotoxicity was significantly reduced by empagliflozin treatment. The apparent discrepancy between cytotoxicity and apoptosis could be explained by the fact that apoptosis is a coordinated and energy-reliant process that involves the activation of caspases, while cell death characterized by loss of cell membrane integrity (which was measured in our cytotoxicity assay) is energy-independent (*Cummings and Schnellmann, 2004*), therefore the two processes can take place independently, sequentially, as well as concurrently (*Elmore, 2007*). Interestingly, we observed a similar trend with regard to LD accumulation. We found a significantly increased number of LDs in AS podocytes compared to WT podocytes, which was reduced by empagliflozin treatment. However, no difference in LD accumulation was observed in tubular cells (*Figure 2*), suggesting that the mechanisms leading to LD accumulation in podocytes in AS are cell-specific. As podocytes but not tubular cells in AS are in contact with an abnormal glomerular basement membrane, the possibility that the LD accumulation is the result of a cross-talk between matrix and lipid metabolism is possible, as it was recently suggested by others (*Romani et al., 2019*). As far as the mechanisms linking a similar trend in LD accumulation and apoptosis, it was reported that lipids such as triglyceride, cholesterol, fatty acids, and ceramide can directly induce caspase activation, leading to programmed cell death (*Huang and Freter, 2015*). The exact mechanism by which lipotoxicity induces apoptosis warrants further investigations.

To test the preference of energy substrate in association with AS and empagliflozin treatment, we measured cellular respiration by high-resolution respirometry (*Figure 3*). After permeabilizing the cells, we sequentially added different substrates and observed their direct effect on oxygen consumption. WT and AS tubular cells consume the same amount of oxygen in the presence of FAO-linked substrates. However, AS tubular cells respire more following the addition of NADH-linked substrates. While not the major source of renal energy, glucose oxidation is crucial in tubular function (*Ross et al., 1986*). Elevated NADH-linked respiration in AS tubular cells may suggest an increased demand for alternative substrates in this cell type under disease conditions. Empagliflozin does not affect tubular cell respiration independently of the substrate used in the assay. However, empagliflozin treatment of AS podocytes inhibits NADH-linked respiration and appears to promote a shift in substrate utilization for ATP production. The accumulation of LD in AS podocytes could lead to an increase in the availability of fatty acids and contribute to the elevated FAO-linked respiration observed in these cells. Given that podocytes rely more on glucose oxidation and are, therefore, more vulnerable to glucose deprivation, it is feasible to speculate that empagliflozin only affects podocyte respiratory metabolism and not that of tubular cells.

To better define the action of SGLT2i, we performed experiments using AS podocytes with siRNA-mediated *Sglt2* knockdown (*Figure 4*). We demonstrated that both siSglt2 and SGLT2i (empagliflozin) similarly reduce apoptosis in AS podocytes. The rate-limiting enzyme of fatty acid oxidation, CPT1A, was upregulated by empagliflozin treatment and siSglt2. It is interesting to note that the mutations in SGLT2 are responsible for familial renal glucosuria (FRG), which is characterized by glucose in the urine (*Santer and Calado, 2010*). The isolation of urine-derived podocytes was previously described (*Sakairi et al., 2010*) and we used urine-derived podocytes from patients to study mitochondrial dysfunction and oxygen consumption (*Ge et al., 2021*). Thus, the study of urinary podocytes from patients with FRG would be a valuable tool to investigate the metabolic switch associated with SGLT2 deficiency and warrants future investigations.

While the major mechanisms by which SGLT2i reduces albuminuria are thought to be linked to a modulation of the tubulo-glomerular feedback resulting in improved glomerular hyperfiltration (*Mabillard and Sayer, 2020*), it is possible that additional mechanisms are also involved. Several studies have demonstrated that SGLT2i has a remarkable effect on lipid metabolism in vivo. For example, SGLT2 inhibition modulates renal lipid metabolism in db/db mice (*Wang et al., 2017*), ameliorates obesity in high-fat diet-fed animal models by improving FAO (*Wei et al., 2020*; *Yokono et al., 2014*), and reduces the cardiotoxic lipids in the hearts of diabetic fatty rats (*Aragón-Herrera et al., 2019*). In this study, we demonstrate that empagliflozin treatment protects against renal disease progression and expands the life span of AS mice. We furthermore demonstrate that empagliflozin not only improves kidney function (*Figure 6*) but also significantly reduces intrarenal lipid accumulation (*Figure 7*) in AS mice. To allow for comparison to the SOC, the ACEi ramipril was also included in our study in order

to determine whether a combination of empagliflozin and ramipril would have a superior effect on ramipril. We show that treatment of AS mice with both ramipril and empagliflozin is not superior in preserving renal function when compared to treatment with ramipril alone. While this is not consistent with the findings reported in patients with DKD, the very sizable effect of ramipril in experimental AS may account for the inability to report additional renoprotective effects of empagliflozin. Interestingly, we found that empagliflozin or the combined treatment of empagliflozin and ramipril reduces triglyceride content in the kidney of AS mice, while ramipril did not have any effect on the renal triglyceride content but affected esterified cholesterol content, suggesting that the effect of ACEi and SGLT2i on renal lipid metabolism may differ. More importantly, we identified a strong correlation between lipid accumulation (CE, triglyceride) in kidney cortices and the decline in renal function (albuminuria, serum BUN and creatinine), which is similar to our previous findings in experimental AS, DKD, and FSGS (*Ducasa et al., 2019*; *Ge et al., 2021*; *Wright et al., 2021*). These data suggest that the renal protection of empagliflozin in AS may at least in part be mediated by its ability to modulate renal lipid metabolism. It is interesting to note that while podocyte-specific glucose transporter (GLUT) 4 deficient podocytes are characterized by morphology change which may be mediated by the lack of nutrients (*Guzman et al., 2014*), *Glut4*-deficient mice are protected from diabetic nephropathy. The beneficial effect of GLUT4 deficiency and SGLT2 inhibition may both be interpreted by the deprivation of nutrients such as glucose. Further studies will be needed to understand the impact of renal lipid content in affected patients and to determine if renoprotection conferred by SGLT2i may be monitored through non-invasive measures of fat content such as Dixon magnetic resonance imaging (*Gaborit et al., 2021*).

We previously reported that one of the drivers of lipotoxic injury in AS is the abnormal production of collagen I using the *Col4a3*$^{-/-}$ model (*Kim et al., 2021*), which has also been reported in *Col4a5*$^{-/-}$ mice (*Randles et al., 2021*). AS is caused by impaired heterotrimerization of α3α4α5 of collagen type IV (mature collagen form) due to any one of the *COL4A3*, *COL4A4*, or *COL4A5* mutations. This results in the persistent production of α1α1α2 of collagen type IV, which is an immature form of the glomerular basement membrane and susceptible to proteinase, during kidney development in AS (*Hudson et al., 2003*). These observations suggest the possibility that some of the mechanisms leading to renal failure in AS may be shared, independently if caused by *COL4A3*, *4*, and *5* mutations. At this time, we also do not know if SGLT2i would be beneficial to protect from lipotoxic injury in other models of AS. Further studies are needed to address the role of lipotoxicity-induced podocyte injury in other forms of AS.

Our study has several limitations. First, we did not study empagliflozin's off-target effect on other transporters such as sodium-hydrogen exchanger (NHE) 1 in the heart or NHE3 in the kidney (*McGuire et al., 2021*). It is possible that these pathways are also involved. However, a recent study shows that empagliflozin does not inhibit NHE1 in the heart (*Chung et al., 2021*), and the way SGLT2i inhibit the NHE isoforms in the kidney remains to be proven (*De Pascalis et al., 2021*). In addition, in vitro experiments were performed using immortalized podocytes and tubular cells established from AS and WT mice. While these cells are very similar to primary cells, they may exhibit some changes in protein expression and function. However, the in vivo study using an experimental model of non-metabolic kidney disease (AS mice) supports our hypothesis. Last, we only focused on podocytes and tubular cells and demonstrated that a metabolic switch occurs in podocytes but not in tubular cells. Nevertheless, we cannot exclude the possibility that other glomerular cells are involved in the substrate switch upon empagliflozin treatment. Therefore, additional analysis of glomerular endothelial cells and mesangial cells is warranted.

In summary, our study demonstrates that empagliflozin reduces podocyte lipotoxicity and improves kidney function in experimental AS. This beneficial effect is associated with a shift in the use of energy substrates from glucose to fatty acids in podocytes. Lipid accumulation in kidney cortices correlates with kidney disease progression, therefore, reducing renal lipid content by empagliflozin and other agents may represent a novel therapeutic strategy for the treatment of patients with AS. Results obtained from this study may allow us to better define the mechanisms leading to SGLT2i-mediated renoprotection in non-diabetic kidney disease.

# Methods

**Key resources table**

| Reagent type (species) or resource | Designation | Source or reference | Identifiers | Additional information |
|---|---|---|---|---|
| Cell line (*M. musculus*) | immortalized podocytes | This paper; PMID:33340991 | | Cell line established and maintained in Fornoni lab |
| Cell line (*M. musculus*) | immortalized tubular cells | This paper | | Cell line established and maintained in Fornoni lab |
| Genetic reagent (*M. musculus*) | CBA/CaxC57BL/10-H-2Kb-tsA58 | Charles River; (PMID:1711218) | | |
| Genetic reagent (*M. musculus*) | 129-*Col4a3*tm1Dec/J | Jackson Laboratory | Strain# 002908 RRID:IMSR_JAX:002908 | |
| Antibody | anti-WT1 (rabbit polyclonal) | Santa Cruz Biotechnology | Cat# sc-192 RRID:AB_632611 | 1:300 (IF) |
| Antibody | anti-SGLT2 (mouse monoclonal) | Santa Cruz Biotechnology | Cat# sc-21537 RRID:AB_2814658 | 1:100 (IHC) 1:500 (WB) |
| Antibody | anti-SGLT2 (rabbit polyclonal) | BiCell scientific | Cat# 20802 RRID:AB_2935905 | 1:100 (IF) |
| Antibody | anti-SYNAPTOPODIN (goat polyclonal) | Santa Cruz Biotechnology | Cat# sc-21537 RRID:AB_2201166 | 1:300 (IF) 1:1,000 (WB) |
| Antibody | anti-AQP1 (rabbit polyclonal) | Proteintech | Cat# 20333–1-AP RRID:AB_10666159 | 1:2,000 (WB) |
| Antibody | anti-CPT1A (mouse monoclonal) | Abcam | Cat# ab128568 RRID:AB_11141632 | 1:1,000 (WB) |
| Antibody | anti-GAPDH (mouse monoclonal) | Sigma-Aldrich | Cat# CB1001 RRID:AB_2107426 | 1:10,000 (WB) |
| Sequence-based reagent | *Sglt2*_F | This paper | PCR primers | ATGGAGCAACACGTAGAGGC |
| Sequence-based reagent | *Sglt2*_R | This paper | PCR primers | ATGACCAGCAGGAAATAGGCA |
| Sequence-based reagent | *Gapdh*_F | This paper | PCR primers | CCTGGAGAAACCTGCCAAGTATG |
| Sequence-based reagent | *Gapdh*_R | This paper | PCR primers | GGTCCTCAGTGTAGCCCAAGATG |
| Sequence-based reagent | siRNA: *Sglt2* | Santa Cruz Biotechnology | Cat# sc-61540 | 20 nM |
| Sequence-based reagent | siRNA: nontargetin control | Thermo Scientific | Cat# 4390843 | 20 nM |
| Commercial kit | ApoTox-Glo Triplex assay | Promega | Cat# G6320 | |
| Commercial kit | Amplex Red Cholesterol Assay | Thermo Scientific | Cat# A12216 | |
| Commercial kit | Triglyceride Colorimetric Assay | Cayman | Cat# 10010303 | |
| Chemical compound, drug | Empagliflozin (BI 10773) | Selleckchem | Cat# S8022 | 500 nM |
| Software, algorithm | Graphpad Prism | Graphpad software | SCR_002798 | |

## Animal studies

### Phenotypic analysis of mice

*Col4a3*−/− mice (a model of AS) are in a 129X1/SvJ background and were purchased from Jackson Laboratory (129-*Col4a3*tm1Dec/J, stock #002908). Mice were fed empagliflozin-supplemented chow (70 mg/kg) versus a regular diet starting at 4 weeks of age. Ramipril was added to the drinking water at a

concentration that would lead to a daily uptake of 10 mg/kg body weight (*Kim et al., 2021*). Five groups of mice were examined: WT + placebo, AS + placebo, AS + empagliflozin, AS + ramipril, and AS + empagliflozin + ramipril. Both male and female mice were used. Mice were sacrificed at 8 weeks and analyzed as described below.

## Urinary albumin-to-creatinine ratios

Morning spot urine samples were collected bi-weekly. Urinary albumin-to-creatinine ratios were determined using the Mouse Albumin ELISA Kit (Bethyl Laboratories, Montgomery, TX) and Creatinine LiquiColor (Stanbio, Boerne, TX). Albuminuria values are expressed as μg albumin per mg creatinine.

## Serology

Blood samples were collected and serum creatinine was determined by tandem mass spectrometry at the UAB-UCSD O'Brian Core Center (University of Alabama at Birmingham) as previously described (*Takahashi et al., 2007*). Serum BUN was analyzed in the Comparative Laboratory Core Facility of the University of Miami.

## Oil red-O staining

Four μm kidney cortices optimal cutting temperature (OCT) compound embedded sections were incubated with 100 μl freshly prepared Oil Red O solution (Electron Microscopy Science, Hatfield, PA) for 15 min and counterstained with Hematoxylin Harris solution (VWR, 10143–606) for 5 min to detect lipid deposition. Images were examined under a light microscope (Olympus BX41, Tokyo, Japan), and quantified by the percentage of LD-positive glomeruli.

## Immunofluorescence staining

To measure podocyte number per glomerulus, glomerular sections embedded in OCT were stained with a Wilms tumor 1 (WT1) antibody (Santa Cruz Biotechnology, Dallas, TX, sc-192, 1:300), followed by a secondary antibody (Invitrogen, Waltham, MA, A-11008, 1:500) and Mounting Medium with DAPI (Vectorlabs, Newark, CA, H-1200). To study the podocyte-specific localization of SGLT2, glomerular sections were stained with podocyte marker SYNPO (Santa Cruz Biotechnology, Dallas, TX, sc-21537, 1:300) and SGLT2 (BiCell scientific, Maryland Heights, MO, 20802, 1:100) with secondary antibodies (Invitrogen, Waltham, MA, A-11055 & A-11036, 1:500). Images were acquired using Olympus IX81 confocal microscope (Tokyo, Japan) coupled with a 60 x oil immersion objective lens and images were processed using Fiji/Image J.

## Kidney histology analysis

Perfused kidneys were fixed in 10% formalin and paraffin-embedded, and then cut into 4 μm thick sections. Periodic acid-Schiff (PAS) staining was performed to investigate mesangial expansion following a standard protocol. The mesangial expansion was visualized under a light microscope (Olympus BX41, Tokyo, Japan) and 20 glomeruli per section were scored by semi-quantitative analysis (scale 0–5), performed in a blinded manner. Picrosirius Red staining was performed to measure fibrosis. Paraffin-embedded sections were deparaffinized with xylene and a graded alcohol series. Sections were rinsed and stained for 1 hr with Picrosirius Red in saturated aqueous picric acid. Sections were examined under a light microscope (Olympus BX41, Tokyo, Japan), followed by analysis with Fiji/Image J.

## Lipid extraction

Kidney cortices were homogenized in a buffer containing 50 mM pH 7.4 potassium phosphate and cOmplete Protease Inhibitor Cocktail tablet (Roche, Indianapolis, IN, 1 pill in 10 ml buffer) by sonication for the 20 s, twice, on ice. Total lipids were extracted from homogenates using hexane:isopropanol (3:2) and placed in a mixer (1000 rpm) for 30 min. The mixed homogenate was then spun at top speed, lipids contained in the supernatants were collected, and pellets were disrupted by 2 sequential lipid extractions. Total lipids were then pooled and dried using a speed vacuum at 37 °C and reconstituted with 100 μl isopropanol:NP-40 (9:1). Proteins were extracted from the pellets using 8 M Urea, 0.1% SDS, and 0.1 M NaOH. Extracted lipids were used for determining total cholesterol, cholesterol ester and triglyceride contents, and normalized to protein concentrations.

## Triglyceride (TG) assay

The TG content was determined using Triglyceride Colorimetric Assay Kit (Cayman, Ann Arbor, MI) following the manufacturer's protocol. TG standards and lipid samples from the above-mentioned extraction were added to a 96-well plate. The reaction was initiated by adding 150 µl enzyme buffer to each well. Absorbance at 540 nm was measured using a SpectraMax M5 plate reader (Molecular Devices, San Jose, CA).

## Cholesterol assay

Cholesterol assays were performed using the Amplex Red Cholesterol Assay Kit (ThermoFisher Scientific, Waltham, MA) following the manufacturer's instructions with some modifications (*Ge et al., 2021*). Total cholesterol and cholesterol ester were quantified using a direct enzymatic method (*Mizoguchi et al., 2004*) and fluorescence was read at 530/580 nm. SpectraMax M5 plate reader (Molecular Devices, San Jose, CA) was used.

## Immunohistochemistry

Four µm kidney sections were heated at 65 °C for 1 hr and deparaffinized in xylene, followed by rehydration in decreasing concentrations of ethanol (two washes in 100% ethanol, two washes in 95%, one wash in 70%, one wash in 50%, and three wash in TBS). Antigen retrieval was performed for 30 min in citrate buffer (Sigma-Aldrich, St. Louis, MO, C9999, 1:10). Sections were incubated with 3% hydrogen peroxidase (Sigma-Aldrich, St. Louis, MO, H1009, 1:10) for 20 min and incubated with a blocking reagent (Vector Laboratories, Newark, CA, SP-5035) for 1 hr at room temperature. Sections were then incubated with primary antibody SGLT2 (Santa Cruz Biotechnology, Dallas, TX, sc-393350, 1:100) overnight at 4 °C. Incubation with biotin-labeled secondary antibody (Vector Laboratories, Newark, CA, BA-2000, 1:200) was performed at room temperature for 1 hr, followed by incubation with avidin-biotin-peroxidase complex (Vector Laboratories, Newark, CA, PK-6100) and DAB substrate kit (Vector Laboratories, Newark, CA, SK-4100). Counterstain was performed with hematoxylin for 5 min, followed by dehydration in increasing concentrations of ethanol. Sections were examined under a light microscope (Olympus BX41, Tokyo, Japan).

## Cell lines

### Establishment and culture of conditionally immortalized mouse podocyte and tubular cell lines

To establish immortalized mouse podocyte and tubular cell lines, *Col4a3*[+/-] mice were bred with the immorto-mice carrying a temperature-sensitive T-antigen transgene (SV40[+]) (Charles River, Wilmington, MA, CBA/CaxC57BL/10-H-2Kb-tsA58) (*Jat et al., 1991*) to generate double heterozygous littermates, which were then crossed to generate SV40[+]; *Col4a3*[-/-] (immorto-AS) and SV40[+]; *Col4a3*[+/+] (immorto-WT) (*Kim et al., 2021*; *Liu et al., 2020*). Glomeruli and tubules were isolated from 9 weeks old immorto-WT and -AS mice by differential sieving as previously described (*Mundel et al., 1997*; *Terryn et al., 2007*). Immortalized cell lines were cultured at 33 °C in RPMI growth medium (containing 10% FBS, 1% penicillin/streptomycin, 100 U/ml IFNγ) under permissive conditions. Podocyte cell lines will be thermo-shifted to 37 °C non-permissive condition in the absence of IFNγ for 12 days. Immortalized mouse podocyte and tubular cell lines were characterized by western blot analysis using podocyte and tubular cell markers. Cultured cells were incubated with 500 nM empagliflozin (Selleckchem, Houston, TX) or dimethylsulfoxide in a growth medium for 48 hr. To furthermore study the effects of SGLT2i, at day 9 of differentiation, AS podocyte cell lines were transfected with *Sglt2* siRNA (20 nM, Santa Cruz Biotechnology, Dallas, TX) or nontargeting siRNA (20 nM, Thermo Scientific, Waltham, MA) for 72 hr using HiPerFect Transfection Reagent (Qiagen, Valencia, CA). After 24 hr of transfection, AS podocytes were then exposed to empagliflozin or vehicle. Cell lines are tested negative for mycoplasma contamination. We have not used any cell lines from the list of commonly misidentified cell lines maintained by the International Cell Line Authentication Committee.

## Quantitative real-time PCR

RNA was extracted from cultured cells using the RNeasy Mini Kit (Qiagen, Valencia, CA). Reverse transcription was performed using qScript cDNA SuperMix (QuantaBio, Beverly, MA). Quantitative real-time PCR was carried out using the StepOnePlus system (Applied Biosystems, Waltham, MA)

with PerfeCTa SYBR Green FastMix (QuantaBio, Beverly, MA). Relative quantification was determined as $2^{-\Delta\Delta Ct}$. The following primers were used: *Sglt2*: forward-ATGGAGCAACACGTAGAGGC, reverse-ATGACCAGCAGGAAATAGGCA; *Gapdh*: forward-CCTGGAGAAACCTGCCAAGTATG, reverse-GGTCCTCAGTGTAGCCCAAGATG. *Sglt2* expressions in Healthy Mouse Dataset were available in the Kidney Interactive Transcriptomics database (https://humphreyslab.com/SingleCell/).

## Western blot analysis

Cell lysates were prepared using 3-[(3-cholamidopropyl)dimethylammonio]–1-propanesulfonic (CHAPS) acid buffer. Protein concentration was measured with the bicinchoninic acid (BCA) reagent (Thermo Scientific, Waltham, MA). 20–30 µg of protein extract was loaded onto 4 to 20% SDS-polyacrylamide gel electrophoresis (SDS-PAGE) gels (Bio-Rad, Hercules, CA) and transferred to Immobilon-P PVDF membranes (Bio-Rad, Hercules, CA). Western blot analysis was performed using a standard protocol and the following primary antibodies: SGLT2 (Santa Cruz Biotechnology, Dallas, TX, sc-393350, 1:500), SYNAPTOPODIN (Santa Cruz Biotechnology, Dallas, TX, sc-21537, 1:1000), AQP1 (Proteintech, Rosemont, IL, 20333–1-AP, 1:2000), CPT1A (Abcam, Cambridge, UK, ab128568, 1:1,000), GAPDH (Sigma-Aldrich, St. Louis, MO, CB1001, 1:10,000); or secondary antibodies: anti-mouse IgG horseradish peroxidase (HRP) (Promega, Madison, WI, W402B, 1:10,000), anti-rabbit IgG HRP (Promega, Madison, WI, W401B, 1:10,000) or anti-goat IgG HRP (Promega, Madison, WI, V805A, 1:10,000). Signal was detected with Radiance ECL (Azure, Dublin, CA) using Azure c600 Imaging System.

## Cytotoxicity and apoptosis assay

Cytotoxicity and apoptosis assays were performed using the ApoTox-Glo Triplex assay (Promega, Madison, WI) according to the manufacturer's protocol. Briefly, mouse tubular cells and differentiated podocytes were cultured and treated as indicated above. Fluorescence was measured at 400 nm excitation/505 nm emission for viability, and 485 nm excitation/520 nm emission for cytotoxicity. Additionally, apoptosis was determined by luminescence for caspase-3/7 activation. Values were expressed as the cytotoxicity/viability and apoptosis/viability ratios, then compared with WT controls. Fluorescence and luminescence were measured on a SpectraMax i3x multi-mode microplate reader (Molecular Devices, San Jose, CA).

## Lipid droplet quantification

Cultured cells were fixed with 4% paraformaldehyde (PFA) and 2% sucrose and then stained with Nile red (Sigma-Aldrich, St. Louis, MO) and High-Content Screening (HCS) Cell Mask Blue (Invitrogen, Waltham, MA) according to the manufacturer's protocols. Images were acquired using the Opera high-content screening system (20 x confocal lens) and lipid droplets intensity per cell was determined using the Columbus Image Analysis System (Perkin Elmer, Waltham, MA) (*Liu et al., 2020*).

## Cellular respiration measurements

Oxygen consumption rate (OCR) was measured using a high-resolution respirometer (O2k-Fluo-Respirometer, Oroboros Instruments, Innsbruck, Austria) filled with 2 mL of mitochondrial respiration buffer (MiR05, containing 0.5 mM EGTA, 3 mM MgCl$_2$·6H$_2$O, 60 mM K-lactobionate, 20 mM Taurine, 10 mM KH$_2$PO$_4$, 20 mM HEPES, 110 mM Sucrose, 1 g/l fatty acid-free BSA) at 37 °C, following the Substrate-Uncoupler-Inhibitor-Titration (SUIT)–002 protocol with some modifications. Specifically, $1 \times 10^6$ suspended cells were immediately placed into the chamber and continuously mixed by a stirrer at 750 rotations per minute. O$_2$ consumption in nearly diffusion-tight closed chambers is calculated in real-time by polarographic oxygen sensors. First, endogenous respiration was measured in intact cells. For substrate-driven respiration, cells were permeabilized with 2.5 ug/ml digitonin (optimal digitonin concentration for podocytes and tubular cells was established prior to following the SUIT-010 protocol) and supplemented with 2.5 mM ADP. FAO-linked substrates (0.5 mM octanoylcarnitine plus 0.1 mM malate) were then added to the chamber using a Hamilton microsyringe, and the coupled FA-driven OCR was measured. Finally, 2 mM malate, 5 mM pyruvate, and 10 mM glutamate were added to initiate coupled NADH-linked respiration, and the additive effect of NADH-driven OCR was measured. Mitochondrial outer membrane integrity was tested by the addition of 10 uM cytochrome *c*. Respiration was inhibited by the addition of 100 mM sodium azide, which is a specific mitochondrial

complex IV (CIV) inhibitor. Cell respiration was recorded as pmol $O_2$ consumed for 1 s and normalized to cell numbers.

## Pyruvate dehydrogenase activity assay

Pyruvate dehydrogenase (PDH) activity in cells was determined using PDH Colorimetric Assay Kit (BioVision, Milpitas, CA) according to the manufacturer's protocol. $1x10^6$ cells were used. Absorbance at 450 nm was measured using a SpectraMax M5 plate reader (Molecular Devices, San Jose, CA).

## Glucose measurement

Glucose contents in cell lysates were quantified using the Glucose-Glo Assay (Promega, Madison, WI), according to the manufacturer's instructions. This assay combines glucose oxidation and NADH generation to produce a luminescence signal proportional to the glucose concentration. Briefly, the cell lysates were incubated with Glucose Detection Reagent for 1 hr at room temperature, then the luminescence was measured by SpectraMax M5 plate reader (Molecular Devices, San Jose, CA).

## Extracellular acidification

Extracellular acidification was determined by Glycolysis Assay [Extracellular Acidification] (Abcam, Cambridge, UK) according to the manufacturer's protocol. Differentiated AS podocytes were cultured in a $CO_2$ incubator and transferred to a $CO_2$-free incubator at 37 °C 3 hr prior to performing the assay. A pH-sensitive reagent was added to detect an increased signal with increased acidification. Fluorescence was measured at 380 nm excitation/615 nm emission on a SpectraMax i3x multi-mode microplate reader (Molecular Devices, San Jose, CA).

## Statistics

For each statistical test, biological sample size (n), and p-value are indicated in the corresponding figure legends. All values are presented as mean ± SD. Statistical analysis was performed using Prism GraphPad 7 software. Significant outliers were determined by the GraphPad outlier calculator and excluded from further statistical analysis. Animals were grouped according to genotypes and then randomized, and investigators were blinded for the analyses. When comparing two groups, a two-tailed Student's t-test was performed. Otherwise, results were analyzed using One-way ANOVA followed by Holm-Sidak's multiple comparisons. A p-value less than 0.05 was considered statistically significant. Only data from independent experiments were analyzed.

## Study approval

All studies involving mice were approved by the Institutional Animal Care and Use Committee (IACUC) at the University of Miami. The University of Miami (UM) has an Animal Welfare Assurance on file with the Office of Laboratory Animal Welfare, NIH (A-3224–01, effective November 24, 2015). Additionally, UM is registered with the US Department of Agriculture Animal and Plant Health Inspection Service, effective December 2014, registration 58 R-007. As of October 22, 2013, the Council on Accreditation of the Association for Assessment and Accreditation of Laboratory Animal Care (AAALAC International) has continued UM's full accreditation.

## Acknowledgements

This project was supported by grants from the National Institutes of Health [grant numbers R01DK117599, R01DK104753, and R01CA227493] to AF and SM, and the Miami Clinical Translational Science Institute [grant numbers U54DK083912, UM1DK100846, U01DK116101, and UL1TR000460] to AF. FF is supported by the Army Research Office [grant number W911NF-21-1-0359]. We thank Dr. Volker Vallon for the kidney lysate from SGLT2$^{-/-}$ mice. A special thanks to the Katz family for supporting this study.

# Additional information

## Competing interests

Kumar Sharma: founder of SygnaMap. Sandra Merscher: is an inventor on pending (PCT/US2019/032215; US 17/057,247; PCT/US2019/041730; PCT/US2013/036484; US 17/259,883; US17/259,883; JP501309/2021, EU19834217.2; CN-201980060078.3; CA2,930,119; CA3,012,773,CA2,852,904) or issued patents (US10,183,038 and US10,052,345) aimed at preventing and treating renal disease. They stand to gain royalties from their future commercialization. SM holds indirect equity interest in, and potential royalty from, ZyVersa Therapeutics, Inc by virtue of assignment and licensure of a patent estate. SM is supported by Aurinia Pharmaceuticals Inc. Alessia Fornoni: is an inventor on pending (PCT/US2019/032215; US 17/057,247; PCT/US2019/041730; PCT/US2013/036484; US 17/259,883; US17/259,883; JP501309/2021, EU19834217.2; CN-201980060078.3; CA2,930,119; CA3,012,773,CA2,852,904) or issued patents (US10,183,038 and US10,052,345) aimed at preventing and treating renal disease. They stand to gain royalties from their future commercialization. AF is Vice-President of L&F Health LLC and is a consultant for ZyVersa Therapeutics, Inc. ZyVersa Therapeutics, Inc has licensed worldwide rights to develop and commercialize hydroxypropyl-beta-cyclodextrin from L&F Research for the treatment of kidney disease. AF also holds equities in Renal 3 River Corporation. AF and SM are supported by Aurinia Pharmaceuticals Inc. The other authors declare that no competing interests exist.

## Funding

| Funder | Grant reference number | Author |
| --- | --- | --- |
| National Institutes of Health | R01DK117599 | Sandra Merscher Alessia Fornoni |
| Miami Clinical and Translational Science Institute, University of Miami | U54DK083912 | Alessia Fornoni |
| Army Research Office | W911NF-21-1-0359 | Flavia Fontanesi |
| National Institutes of Health | R01CA227493 | Sandra Merscher Alessia Fornoni |
| National Institutes of Health | R01DK104753 | Sandra Merscher Alessia Fornoni |
| Miami Clinical and Translational Science Institute, University of Miami | UL1TR000460 | Alessia Fornoni |
| Miami Clinical and Translational Science Institute, University of Miami | U01DK116101 | Alessia Fornoni |
| Miami Clinical and Translational Science Institute, University of Miami | UM1DK100846 | Alessia Fornoni |

The funders had no role in study design, data collection and interpretation, or the decision to submit the work for publication.

## Author contributions

Mengyuan Ge, Data curation, Formal analysis, Investigation, Methodology, Writing - original draft, Writing – review and editing; Judith Molina, Data curation, Formal analysis, Investigation, Methodology; Jin-Ju Kim, Data curation, Investigation, Methodology; Shamroop K Mallela, Anis Ahmad, Alla Mitrofanova, Investigation; Javier Varona Santos, Hassan Al-Ali, Methodology; Kumar Sharma, Investigation, Writing – review and editing; Flavia Fontanesi, Formal analysis, Supervision, Validation, Investigation, Methodology, Writing – review and editing; Sandra Merscher, Formal analysis, Supervision,

Funding acquisition, Validation, Methodology, Writing – review and editing; Alessia Fornoni, Conceptualization, Formal analysis, Supervision, Funding acquisition, Validation, Methodology, Project administration, Writing – review and editing

## Author ORCIDs
Mengyuan Ge (ID) http://orcid.org/0000-0002-3735-377X
Alessia Fornoni (ID) http://orcid.org/0000-0002-1313-7773

## Ethics
All studies involving mice were approved by the Institutional Animal Care and Use Committee (IACUC) at the University of Miami. The University of Miami (UM) has an Animal Welfare Assurance on file with the Office of Laboratory Animal Welfare, NIH (A-3224-01, effective November 24, 2015). Additionally, UM is registered with the US Department of Agriculture Animal and Plant Health Inspection Service, effective December 2014, registration 58-R-007. As of October 22, 2013, the Council on Accreditation of the Association for Assessment and Accreditation of Laboratory Animal Care (AAALAC International) has continued UM's full accreditation.

## Decision letter and Author response
Decision letter https://doi.org/10.7554/eLife.83353.sa1
Author response https://doi.org/10.7554/eLife.83353.sa2

## Additional files

### Supplementary files
• MDAR checklist
• Source data 1. Full blot images of all western blots.

### Data availability
All data generated or analysed during this study are included in the manuscript and supporting file.

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
