## [Editor Report]

The article has important scientific merit in the field of glomerular diseases. The authors propose a link between inhibition of SGLT2 and lipotoxicity-mediated renal injury in experimental Alport syndrome (AS) by modulation pathways linked to CKD progression, possibly through metabolic adaption in podocytes.

---

## [Decision Letter]

**Decision letter after peer review:**

Thank you for submitting your article "Empagliflozin reduces renal lipotoxicity in experimental Alport syndrome" for consideration by *eLife*. Your article has been reviewed by 3 peer reviewers, one of whom is a member of our Board of Reviewing Editors, and the evaluation has been overseen by Martin Pollak as the Senior Editor. The following individual involved in review of your submission has agreed to reveal their identity: John A Sayer (Reviewer #3).

Essential revisions:

1) Authors show SGLT2 expression in cultured podocytes, and that glucose/pyruvate handling was different in podocytes resulting in reduced lipotoxicity with SGLT2i treatment is an interesting finding. However whether the mechanism is exclusively through the podocytes, authors do not yet answer this intriguing question. Experiments are needed to clarify podocyte SGLT2 expression in AS in vivo and whether limiting SGLT2 exclusively in podocyte is important in AS, or whether other glomerular cells could be at play.

2) As stated by Reviewer 1, experiments with knockout SGLT2 in podocytes are needed to define the actions of empagliflozin in podocytes.

3) Details of substrate culture media should be provided

*Reviewer #1 (Recommendations for the authors):*

1. Figure 1 shows human glomerular expression for SGLT2, however the staining is not exclusive to podocyte. These results can be further strengthened by showing either human AS patient biopsy showing SGLT2 expression colocalized with a podocyte marker to justify the focus on podocytes. Include clinical information on patient biopsy. Alternatively show glomerular SGLT2 expression in mice.

2. The authors should include mRNA expression in WT and AS cells. This will ensure that the pathway under study is engaged in the experimental model.

3. There is a lot of literature that supports the pleiotropic effects of SGLT2i effects on glomerular cells including endothelial and mesangial cells. Authors should acknowledge this body of evidence and justify the focus of their study on podocytes in the introduction and/or Discussion section.

4. Figure 2 B,D,G apoptosis measurement are ambiguous. Cell death should be shown as a percentage. As it stands it is misleading when looking at the scales and comparing effects between tubular cell and podocytes.

5. Please provide information about tubular cells and podocytes in cytotoxicity and apoptosis assays. Are the cell tested fully differentiated after thermo-switching? If so, how long? What's in the culture medium? How do AS podocyte survive in permissive condition (i.e. 33 degrees C)? This is important as it would help interpret whether culture conditions impact the cell's state, especially as the cell are dyeing no doubt their metabolic state would be impacted.

6. An important experiment that the authors should consider is to show how much of the effect of empagliflozin is through SGLT2? Anti-lipotoxic effects of SGLT2i should be studied after a genetic or siRNA mediated knock-down of SGLT2 in AS podocytes under conditions of empagliflozin treatment.

7. Did the acidification rate go up in AS podocytes treated with empagliflozin?

*Reviewer #2 (Recommendations for the authors):*

1. 'Podocytes rely more on glucose oxidation and are therefore more vulnerable to glucose deprivation'. Have the authors conducted any experiments to determine the glucose content of cells and/or culture medium following empagliflozin treatment?

2. Figure 6, E-J. The shape of the groups is difficult to discern. Consider labeling the symbols in larger sizes or different colors.

*Reviewer #3 (Recommendations for the authors):*

An analysis of GTEX and other RNA databases for SGLT2 expression in podocytes would be interesting.

Consideration of patients with renal glycosuria and how this could be used to strengthen results.

Some consideration of whether Col4a5 would have exact mechanism of disease and be amenable for treatment.

---

## [Author Response]

Essential revisions:1) Authors show SGLT2 expression in cultured podocytes, and that glucose/pyruvate handling was different in podocytes resulting in reduced lipotoxicity with SGLT2i treatment is an interesting finding. However whether the mechanism is exclusively through the podocytes, authors do not yet answer this intriguing question. Experiments are needed to clarify podocyte SGLT2 expression in AS in vivo and whether limiting SGLT2 exclusively in podocyte is important in AS, or whether other glomerular cells could be at play.

We thank the reviewer for this important comment. We have now included the immunofluorescence staining of kidney sections of Alport mice in Figure 1F, which indicates that SGLT2 (red) colocalizes with the podocyte marker Synaptopodin (green) suggesting SGLT2 expression in podocytes of Alport mice.

We also performed siRNA knockdown of Sglt2 in AS podocytes, in order to study the specific effect of SGLT2 inhibition in podocytes. We show that siSglt2-transfected AS podocytes have reduced apoptosis and increased expression of CPT1A protein levels compared to nontargeting siRNA-transfected AS podocytes, suggesting the knockdown of Sglt2 reduces lipotoxicity (Figure 4). These observations are similar to what we found in SGLT2i-treated AS podocytes. However, we cannot rule the possibility out that other glomerular cells also contribute to reducing lipotoxicity. We have added these limitations in the discussion.

2) As stated by Reviewer 1, experiments with knockout SGLT2 in podocytes are needed to define the actions of empagliflozin in podocytes.

We analyzed the effect of SGLT2 inhibition in Sglt2 siRNA (siSglt2) and nontargeting siRNA (siCtrl) on apoptosis in AS podocytes that were treated with empagliflozin or vehicle and found that the anti-lipotoxic effect of siSglt2 is similar to empagliflozin treatment. These data were implemented in Figure 4.

3) Details of substrate culture media should be provided

We now provide the details of the cell culture media in the methods section (on Page 21). Please also see the answers to Reviewer 1, Question 4.

Reviewer #1 (Recommendations for the authors):1. Figure 1 shows human glomerular expression for SGLT2, however the staining is not exclusive to podocyte. These results can be further strengthened by showing either human AS patient biopsy showing SGLT2 expression colocalized with a podocyte marker to justify the focus on podocytes. Include clinical information on patient biopsy. Alternatively show glomerular SGLT2 expression in mice.

We greatly appreciate the reviewer’s comment on the glomerular expression of SGLT2. To strengthen our results, we performed immunofluorescence (IF) staining using kidney section slides from AS mice. The IF staining image (Figure 1F) validates the colocalization of SGLT2 and the glomerular marker Synaptopodin (SYNPO). Additionally, patient biopsies are de-identified cadaveric biopsies obtained from donors with research consent for which no demographic and clinical informations were made available to us.

2. The authors should include mRNA expression in WT and AS cells. This will ensure that the pathway under study is engaged in the experimental model.

Thank you for your valuable advice. We now added additional results showing Sglt2 mRNA expression levels in WT and AS cells. These data are shown in Figure 1E.

3. There is a lot of literature that supports the pleiotropic effects of SGLT2i effects on glomerular cells including endothelial and mesangial cells. Authors should acknowledge this body of evidence and justify the focus of their study on podocytes in the introduction and/or Discussion section.

We thank the reviewer for bringing up this important point. We added the relevant studies on glomerular endothelial and mesangial cells in the Discussion section and discuss this point as a limitation of our study. We also highlighted that the focus of our study was on podocytes and that this study was aimed at comparing the substrate switch effect in podocytes and tubular cells.

4. Figure 2 B,D,G apoptosis measurement are ambiguous. Cell death should be shown as a percentage. As it stands it is misleading when looking at the scales and comparing effects between tubular cell and podocytes.

We thank the reviewer for the suggestion. We performed the apoptosis assay using the ApoTox-Glo triplex Assay kit, which assesses cell viability, cytotoxicity and caspase activation events (apoptosis) within a single well. This assay allows the normalization of cytotoxicity and apoptosis to viability, with the data presented as the cytotoxicity/viability and apoptosis/viability. To better represent the data, all normalized cytotoxicity and apoptosis data were compared to the WT group. The methods section on page 22 was modified accordingly.

5. Please provide information about tubular cells and podocytes in cytotoxicity and apoptosis assays. Are the cell tested fully differentiated after thermo-switching? If so, how long? What's in the culture medium? How do AS podocyte survive in permissive condition (i.e. 33 degrees C)? This is important as it would help interpret whether culture conditions impact the cell's state, especially as the cell are dyeing no doubt their metabolic state would be impacted.

We thank the reviewer for this comment. Both immortalized podocytes and tubular cells were cultured in RPMI medium (containing 10% FBS, 1%penicillin/streptomycin, 100 U/ml IFNγ) at 33°C during proliferation stage. For podocytes, we performed cytotoxicity and apoptosis assays (and all the other functional assays) 12 days after thermo-switching in the absence of IFNγ. AS podocytes can be propagated at 33 °C and then be thermoshifed to 37°C for proliferation. Synaptopodin (SYNPO) expression was confirmed by western blot analysis in podocytes differentiated for 12 days. Tubular cells were only cultured under permissive condition. AQP1, a tubular cell marker, expression was also verified. These results are now shown in Figure 1 – —figure supplement 1.

6. An important experiment that the authors should consider is to show how much of the effect of empagliflozin is through SGLT2? Anti-lipotoxic effects of SGLT2i should be studied after a genetic or siRNA mediated knock-down of SGLT2 in AS podocytes under conditions of empagliflozin treatment.

We thank the reviewer for the suggestion and took the criticism to heart. As suggested, we performed siRNA knockdown of Sglt2 (siSglt2) in AS podocytes, and used nontargeting siRNA as a control (siCtrl). Cell cytotoxicity and apoptosis of AS podocytes were analyzed in siCtrl, siSglt2, siCtrl+empa, and siSglt2+empa groups. Similar to SGLT2i-treated control AS podocytes (siCtrl+empa), siSglt2 AS podocytes show reduced apoptosis compared to siCtrl (Figure 4). To study fatty acid oxidation (FAO) in siSglt2 AS podocytes, carnitine palmitoyltransferase 1A (CPT1A) was determined by western blot analysis. CPT1A is the rate-limiting enzyme of FAO (Schlaepfer and Joshi, 2020), which was found upregulated by empagliflozin treatment and knockdown of Sglt2 (Figure 4).

7. Did the acidification rate go up in AS podocytes treated with empagliflozin?

Thank you for the insightful question. We investigated acidification rate using Glycolysis Assay [Extracellular Acidification] (Abcam), which detects a change of the fluorescence signal by a pH-sensitive reagent. An increased signal is hereby associated with increased acidification. We found that empagliflozin treatment reduces the acidification rate in AS podocytes (Figure 3—figure supplement 1C). This finding is similar to what was observed in diabetic kidneys (Li et al., 2020) where aberrant glycolysis was inhibited by empagliflozin.

Reviewer #2 (Recommendations for the authors):1. 'Podocytes rely more on glucose oxidation and are therefore more vulnerable to glucose deprivation'. Have the authors conducted any experiments to determine the glucose content of cells and/or culture medium following empagliflozin treatment?

We greatly appreciate the reviewer’s comment. We determined the glucose content in cell lysates using Glucose-Glo Assay kit (Promega). We show that treatment with empagliflozin significantly reduces the glucose content in both podocytes and tubular cells (Figure 2 – —figure supplement 1).

2. Figure 6, E-J. The shape of the groups is difficult to discern. Consider labeling the symbols in larger sizes or different colors.

We thank the reviewer for the suggestion. We now use color figures to allow the easy identification of the different groups.

Reviewer #3 (Recommendations for the authors):An analysis of GTEX and other RNA databases for SGLT2 expression in podocytes would be interesting.

We thank the reviewer for the suggestion. We examined the Kidney Interactive Transcriptomics (https://humphreyslab.com/SingleCell/) which is an online analysis tool for kidney single cell datasets.

In the Healthy Mouse Dataset (Wu, Kirita, Donnelly, and Humphreys, 2019), Sglt2 is expressed in podocytes, though its expression level in every cell type is low.

Consideration of patients with renal glycosuria and how this could be used to strengthen results.

We thank the reviewer for raising this interesting question. It would be important to establish urinary podocytes from patients with renal glycosuria, and we plan to include them in future studies. With regard to the current manuscript, we have included this point in the discussion on page 12.

It is interesting to note that the mutations in SGLT2 are responsible for familial renal glucosuria (FRG), which is characterized by glucose in the urine (Santer and Calado, 2010). The isolation of urine-derived podocytes was previously described (Sakairi et al., 2010) and we used urine-derived podocytes from patients to study mitochondrial dysfunction and oxygen consumption (Ge et al., 2021). Thus, the study of urinary podocytes from patients with FRG would be a valuable tool to investigate the metabolic switch associated with SGLT2 deficiency and warrants future investigations.

Some consideration of whether Col4a5 would have exact mechanism of disease and be amenable for treatment.

We thank the reviewer for raising this important point and we have included this consideration in our discussion. While we have not investigated if lipotoxicity-mediated podocyte injury also plays a role in other forms of experimental AS, such as Col4a5 deficiency, we previously reported that one of the drivers of lipotoxic injury in AS is the abnormal production of collagen I using the Col4a3^-/-^ model (Kim et al., 2021), which has also been reported in Col4a5^-/-^mice (Randles et al., 2021). AS is caused by impaired heterotrimerization of α3α4α5 of collagen type IV (mature collagen form) due to any one of the Col4a3, Col4a4, or Col4a5 mutations. This results in the persistent production of α1α1α2 of collagen type IV, which is an immature form of the glomerular basement membrane and susceptible to proteinase, during kidney development in AS (Hudson, Tryggvason, Sundaramoorthy, and Neilson, 2003). These observations suggest the possibility that some of the mechanisms leading to renal failure in AS may be shared, independently if caused by Col4a3, 4, and 5 mutations. At this time, we also do not know if SGLT2i would be beneficial to protect from lipotoxic injury in other models of AS. Further studies are needed to address the role of lipotoxicity-induced podocyte injury in other forms of AS.